# Effects of Mulch and Fertilization on the Quantity and Quality of Perennial Wall–Rocket (*Diplotaxis tenuifolia*)

**DOI:** 10.3390/plants14101421

**Published:** 2025-05-09

**Authors:** Cristina Precupeanu, Georgiana Rădeanu, Gabriel-Ciprian Teliban, Mihaela Roșca, José Luis Ordóñez-Díaz, Jose Manuel Moreno-Rojas, Vasile Stoleru

**Affiliations:** 1Department of Horticultural Technologies, Faculty of Horticulture, “Ion Ionescu de la Brad” Iasi University of Life Sciences, 3 M. Sadoveanu Alley, 700490 Iasi, Romania; cristina_precupeanu@yahoo.com (C.P.); georgiana_20097@yahoo.com (G.R.); gabriel.teliban@iuls.ro (G.-C.T.); mihaela.rosca@iuls.ro (M.R.); 2Department of Agroindustry and Food Quality, Andalusian Institute of Agricultural and Fisheries Research and Training (IFAPA), Alameda del Obispo, Avda. Menéndez Pidal, SN, 14004 Córdoba, Spain; josel.ordonez@juntadeandalucia.es (J.L.O.-D.); josem.moreno.rojas@juntadeandalucia.es (J.M.M.-R.)

**Keywords:** yield, tunnel crops, food diversity, nutrition quality, fertilization practices

## Abstract

*Diplotaxis tenuifolia*, a species with high nutritional value, was recently introduced in Romania, making in-depth research necessary to develop an efficient cultivation technology to increase agronomic and economic potential. Therefore, the present study aimed to evaluate the influence of three mulch treatments—white polyethylene film (WLDPE), black polyethylene film (BLDPE), and nonmulched (NM)—along with three fertilization regimes—organic (OF), chemical (ChF), and nonfertilized (NF)—on the yield and quality of the Bologna cultivar of perennial wall–rocket under the climatic conditions of northeastern Romania. The results showed that mulching with white polyethylene films significantly increased the CO_2_ assimilation rate, although it did not lead to substantial differences in agro-morphological traits compared to the non-mulched variant. However, plants grown under WLDPE exhibited a significantly higher leaf area index and yield than those under BLDPE mulch. In contrast, BLDPE mulch had a positive effect on dry matter accumulation and β-carotene content. The variations in fertilization regime had no significant impact on most traits analyzed. Significant differences were noted in the CO_2_ assimilation rate and DPPH antioxidant activity, with organic fertilization increasing CO_2_ assimilation and decreasing DPPH activity compared to chemical and unfertilized regimes. Furthermore, the interaction between mulching practices and fertilization regimes revealed significant influences on the physiological performance and phytochemical composition of perennial wall–rocket. The highest CO_2_ assimilation rate and lowest antioxidant activity were recorded in the WLDPE × OF combination, suggesting improved photosynthetic efficiency and a reduced oxidative response resulting from the synergistic effects of reflective mulching and organic fertilization. In contrast, the Bologna cultivar experienced the greatest oxidative stress under the unfertilized regime, with the most pronounced effects observed under no mulching.

## 1. Introduction

*Diplotaxis tenuifolia* L. is a functional food because of the direct connection between its consumption and health benefits [1]. The leaves of this species have a recognizable taste [2] and are distinguished by their dual characteristics, being both a nutritious food and a remedy with therapeutic potential [3]. Rich in glucosinolates, vitamin C, and polyphenol contents, perennial wall rockets are increasingly valued for their antioxidant properties and potential role in protecting against cancer and heart disease [4,5]. Originally present in wild flora, *Diplotaxis tenuifolia* is gaining particular importance [6] in the context of conservation and environmental protection because of the sustainable cultivation technologies associated with it [7]. Perennial wall–rocket is a long-day plant that prefers relatively low temperatures, and relatively high temperatures can stress plants, leading to accelerated flowering and inhibited vegetative growth [8,9,10]. Caruso et al. [6] reported faster growth and development rates in the 2–25 °C temperature range. Moreover, this species can grow in harsh and nutrient-poor soils in its natural habitat [11,12] and has low to moderate nutritional requirements [6]. Moreover, it is a crop with multiple growing cycles, allowing successive harvests that maximize yield and optimize resource consumption [13].

Proper fertilization and mulching practices are essential for supporting the rapid growth and overall health of leafy vegetables, ultimately optimizing yield at harvest [14,15,16,17,18]. The selection of the appropriate type of mulch and fertilizer depends on the specific needs of the plant as well as the climatic conditions of the growing area. In the case of perennial wall rocket, successful cultivation is highly influenced by both environmental factors and the agricultural techniques employed [19]. Although many studies have been performed on *Eruca sativa* [20], the optimal cultivation technologies for this species do not necessarily apply to *Diplotaxis tenuifolia.* These two species can respond differently to the same growing conditions [21,22,23]. Therefore, further studies are needed to establish the best cultivation practices for perennial wall–rocket. As highlighted by Guijarro-Real [24], careful analysis of cultivation techniques is crucial for converting edible wild plants into crops. This analysis is essential for achieving efficient large-scale production while maintaining high quality, including nutritional value. In this context, testing various growing conditions can help identify the most suitable environments for the cultivation of perennial wall–rocket.

Mulch film plays an important role in reducing soil water evaporation [25], improving the soil microclimate, increasing the soil temperature, and stopping weed growth [26]. Thus, its use is highly recommended for most vegetable crops, especially leafy vegetables [27], as a water shortage can significantly reduce yields by decreasing both the number and size of leaves per plant [28]. In perennial wall–rocket crops, insufficient water negatively impacts quality, including its antioxidant content [29]. The mulching film plays a protective role by covering the soil and helping to create a local microclimate around the base of the plant. Furthermore, it helps maintain optimal temperature and humidity levels, which in turn promotes crop growth [30]. Another aspect that strongly impacts the growth and development of plants is the color of the mulching film [21]. Various types of mulching films are available for vegetable cultivation, with black mulching film being the most common [31]. However, other colors, such as brown, transparent, and white, are also utilized [32].

The production of leafy vegetables is also highly dependent on the fertilization technology used [16,20]. It is essential to manage the use of fertilizers carefully for plants with short growth cycles, as these plants exhibit rapid growth rates and high nutrient requirements. These plants can accumulate excessive amounts of nutrients, increasing their vulnerability to disease and posing risks to human health and the environment [4,33]. For example, Tallarita et al. [4] reported that a perennial wall–rocket is a hyperaccumulator of nitrates that is capable of storing up to 9000 mg kg^−1^ f.w. or more, according to growth conditions, often exceeding the maximum limits established by European legislation. Another important aspect of fertilization is the choice of fertilizer type. Some studies have shown that, over the long term, the effectiveness of chemical fertilizers has diminished [34], which has led farmers to increasingly explore the use of organic fertilizers.

Perennial wall–rocket, native to the Mediterranean and western Asia [11,35], can be affected by cold winters and the high humidity of Romania’s climate, and both protective measures and studies are needed to establish optimal growing conditions. Research on this species in Romania is quite limited. One notable study conducted by Teliban et al. [36] evaluated the effects of mulching, along with chemical and biological fertilization, on the growth of perennial wall–rocket during September and October. The researcher reported that the highest yield of 25.3 t·ha^−1^ was achieved in the version using chemical fertilization with white polyethylene film mulch. Instead, Precupeanu et al. [37] investigated the growth and development of the cultivar Seledor in the spring–summer climatic conditions of Romania’s northeast region. This research examined the effects of chemical and organic fertilization, as well as the use of white and black film mulching. The findings of this study indicated that the highest yields were associated with organic fertilization and mulching with white polyethylene film. However, they also noted that the increase in ambient temperature during the spring–summer season led to a decrease in yield. The seasonal dependence of yield was further highlighted in a study conducted by Caruso et al. [38], who reported that the highest production levels of the cultivar Nature occurred during the spring cycle (13.2 t·ha^−1^), followed by the autumn–winter cycle (11.9 t·ha^−1^) and the winter cycle (11.0 t·ha^−1^). Consequently, a more in-depth study on different perennial wall–rocket cultivars is necessary to validate previous findings and determine the most suitable growing season under Romania’s climatic conditions.

The type of mulching and fertilization regime can significantly impact the synthesis of both assimilatory pigments, such as chlorophyll a and b, lycopene, and carotenoids, as well as secondary metabolites like phenolic compounds and flavonoids [38,39,40], compounds that offer significant benefits both to plant health and in human nutrition [41,42]. According to the results of Caruso et al.’s [38] study, the variation in mulching type led to significant differences in the content of polyphenols, ascorbic acid, lipophilic antioxidant activity, hydrophilic antioxidant activity, or color components of perennial wall rocket. Similarly, Franquera [40] found that the levels of chlorophyll a and b in lettuce were significantly higher under yellow and red plastic mulch compared to silver, orange, and green mulch. In contrast, the red plastic mulch induced the highest content of total soluble solids, while the yellow plastic mulch led to the highest sugar content in lettuce. Regarding the fertilization regime, Keçe et al. [43] reported that the chlorophyll contents in lettuce leaves were substantially higher under the influence of organic and inorganic fertilization compared to those detected in the non-fertilized variant. Instead, the study by da Cruz Bento et al. [44] showed that lettuce fertilized with conventional mineral fertilizer, tanned cattle manure, or Geofert organic-mineral fertilizer had lower chlorophyll a but higher chlorophyll b content than non-fertilized plants.

In this context, this study aimed to evaluate the individual and combined effects of white polyethylene film (WLDPE), black polyethylene film (BLDPE), and non-mulched (NM) practices, in conjunction with organic (OF), chemical (ChF), and unfertilized fertilization (NF) regimes, on the yield and quality of the Bologna cultivar of perennial wall–rocket under the climatic conditions of northeastern Romania. To identify the optimal combination of factors for perennial wall–rocket, this study measured parameters such as CO_2_ assimilation rate, the number of leaves, the leaf area index per nest, yield, dry matter and water content, antioxidant activities (measured via DPPH and ABTS assays), total phenolic content (TPC), tannin, chlorophyll a and b, lycopene, and β-carotene levels.

## 2. Results

### 2.1. Individual Influences of Factors on the Characteristics of Perennial Wall–Rocket

Measurements of photosynthetic activity in perennial wall–rocket plants revealed that different fertilization regimes and mulching practices significantly influence plant growth. As shown in Figure 1, plants grown with WLDPE mulch exhibited a significantly higher rate of carbon dioxide assimilation (13.69 µmol m^−2^ s^−1^) compared to those grown without mulch, although the difference was not significant when compared to the BLDPE variant. Among fertilization treatments, only the organic regime significantly enhanced the assimilation rate, reaching 14.38 µmol m^−2^ s^−1^, whereas the chemical and unfertilized variants showed similar, lower values.

The evaluation of fertilization and mulching practices by counting the number of leaves per nest and measuring the biomass, leaf area, and total harvest revealed that the fertilization regimes did not induce significant changes in any of the analyzed parameters. In contrast, the mulching practices significantly affected both the leaf area index (LAI) and the yield of perennial wall–rocket. Although the number of leaves ranged from 557.33 to 660.67 under WLDPE mulch, indicating a difference of up to 18.50% compared to the other two mulching methods, this variation was not statistically significant. However, the LAI for leaves grown under WLDPE mulch (11,309.75 cm^2^·cm^−2^) was significantly higher than that observed for the BLDPE mulch variant (9481.45 cm^2^·cm^−2^). Furthermore, by correlating the LAI values with the number of leaves, it was found that a single leaf from the NM variant exhibited a larger mean LAI (17.19 cm^2^·cm^−2^) than a leaf from the BLDPE mulch variants (16.42 cm^2^·cm^−2^) (Table 1).

In terms of yield, the 45.54 t·ha^−1^ obtained under the influence of the BLDPE mulch was significantly lower than that under the WLDPE mulch (56.36 t·ha^−1^), but insignificantly compared to the yield of the NM variant. Even if the fertilization regime did not significantly influence the perennial wall–rocket yield, the organic fertilization regime effectively increased production, leading to a yield increase of 7.71 t·ha^−1^.

The harvest dynamics presented in Figure 2 show that variation in the type of mulch or fertilization regime had no significant influence on the yield of *Diplotaxis tenuifolia* in the first three harvests but induced significant differences in the fourth. Specifically, the yield under the BLDPE mulch was significantly lower than that under the WLDPE mulch, and the yield under chemical fertilization was significantly lower than those obtained under organic fertilization or no fertilization. Moreover, from the data presented, it can be observed that the highest yields are obtained at the first two harvests, regardless of treatment, but they decrease significantly at the third and fourth harvests, indicating that the growing period of perennial wall–rocket significantly influences the yield.

However, the highest yields were obtained under the influence of white mulch film or organic fertilization, suggesting that these treatments are the most suitable for enhancing the growth of perennial wall–rocket.

In terms of quality analysis, the results obtained revealed that only variations in mulching practices significantly impacted leaf dry weight (Table 2). The dry matter content under BLDPE mulch (8.95%) was significantly higher than in the other two treatments.

Regarding total phenolic content (TPC) and tannin levels, fertilization and mulching practices did not significantly affect their concentrations in perennial wall–rocket leaves (Table 2). However, the non-mulched and non-fertilized treatments were associated with the highest TPC values (2.04 mg GAE·100 g^−1^ and 2.07 mg GAE·100 g^−1^, respectively).

The analysis of antioxidant activity using the ABTS assay indicated that variations in mulching practices and fertilization regimes did not lead to significant changes in the content of antioxidant compounds (Figure 3). In contrast, the DPPH assay revealed that only variations in the fertilization regime significantly affected antioxidant levels, with the highest values recorded under the unfertilized regime (0.49 mmol TE·100 g^−1^ d.w.), suggesting that this treatment induced the highest stress in the perennial wall–rocket. However, antioxidant activity under chemical fertilization was slightly lower but comparable (0.46 mmol TE-100 g^−1^ d.w.), indicating a reduction in stress compared to the NF treatment.

Regarding chlorophyll a, chlorophyll b, and lycopene content, variations in mulch type or fertilization regime did not exert a significant individual effect. According to the data presented in Table 3, significant differences were observed only for β-carotene, with BLDPE mulch favoring the synthesis of this compound (5.99 mg·100 g^−1^ d.w.). In contrast, no significant differences in β-carotene content were observed among the fertilization variants. However, the BLDPE mulch induced the highest levels of chlorophyll a, chlorophyll b, and β-carotene, while the WLDPE mulch resulted in the highest concentration of lycopene. Conversely, organic fertilization led to the highest concentrations of chlorophyll a (94.92 mg·100 g^−1^ d.w.) and lycopene (9.73 mg·100 g^−1^ d.w.), whereas the control treatment presented the highest values for chlorophyll b (39.18 mg·100 g^−1^ d.w.) and β-carotene (5.36 mg·100 g^−1^ d.w.).

### 2.2. Cumulative Influence of Factors on the Characteristics of Perennial Wall–Rocket

The data presented in Figure 4 highlighted that the interaction effects between factors significantly affected assimilation rates in perennial wall–rocket leaves. Thus, the combination of BLDPE mulch with chemical fertilization or of nonmulching practice with a non-fertilization regime induced a significant suppression of CO_2_ assimilation rates compared to the other factor combinations. In contrast, the highest rates were observed for the WLDPE × OF, WLDPE × ChF, BLDPE × NF, and BLDPE × OF variants, ranging from 14.5 µmol m^−2^ s^−1^ to 15.07 µmol m^−2^ s^−1^, with the WLDPE × ChF recording the highest value. However, no statistically significant differences were detected among these variants.

In addition, as shown in Table 4, specific combinations of experimental factors also led to statistically significant variations in agromorphological traits. For example, the number of leaves quantified in the BLDPE × ChF variant, as well as their LAI, were significantly lower only compared to the values associated with the WLDPE × OF, WLDPE × ChF, WLDPE × NF, and NM × OF treatments. With regard to yield, statistically significant differences were observed among the same experimental variants, further highlighting the influence of specific treatment combinations on crop performance. As indicated in Table 4, the highest yields were obtained under the NM × OF, WLDPE × OF, and WLDPE × ChF treatments.

The influence of the combination of experimental factors on the yield of perennial wall–rocket varies depending on the growing period, as illustrated in Figure 5. Thus, at the first harvest, the highest yield (23.74 t·ha^−1^) was recorded in the nonmulched variant with organic fertilization, which was significantly higher than the yields obtained in the WLDPE × OF, WLDPE × ChF, BLDPE × ChF, and NM × ChF treatments. In contrast, at the second harvest, the highest yield (23.41 t·ha^−1^) was recorded in the organically fertilized variant mulched with white film, which was closely followed by the WLDPE × ChF variant (22.23 t·ha^−1^) and was significantly different compared only to the yield harvest from BLDPE × OF and BLDPE × NF. At the third harvest, since the highest yield (12.70 t·ha^−1^) resulted from the combination of white film mulching with chemical fertilization, the differences between experimental variants were insignificant. Instead, the results from the fourth harvest clearly show that the BLDPE × ChF combination resulted in the greatest reduction in perennial wall–rocket yield. The value associated with this experimental variant was significantly lower than those of the other experimental variants, with the exception of the value for NM × ChF.

The data in Table 5 indicate that the combination of the factors had no significant effect on the total polyphenol content but led to significant differences in dry matter content and tannin content. With respect to the dry weight, the results revealed that the BLDPE × ChF plants presented the highest dry weight, a value that is statistically significantly higher than the values recorded for the WLDPE × OF, WLDPE × ChF, WLDPE × NF, BLDPE × NF, NM × OF, and NM × NF variants. In contrast, the lowest dry weight recorded in the WLDPE × ChF variant was significantly lower compared to those of WLDPE × NF, BLDPE × OF, BLDPE × ChF, BLDPE × NF, and NM × ChF.

In terms of tannin content, plants grown under the NM × NF combination exhibited the highest levels, which were statistically significantly different only from those associated with the NM × OF and BLDPE × ChF combinations, which had the lowest values. Therefore, in general, the combination of factors does not exert significant effects on tannin content.

The results of the antioxidant activity analyses presented in Figure 6 indicated that the highest activity, as determined by the DPPH test, was detected in BLDPE × ChF and NM × NF variants, values that were significantly higher than those detected in the other variants (0.54 mmol TE 100 g−1 d.w.). In contrast, the lowest value was recorded for the interaction between non-mulched and organic fertilization (0.43 mmol TE 100 g^−1^ d.w.). Concerning the antioxidant activity assessed through the ABTS test, the findings revealed that the highest value for the WLDPE × ChF variant (0.50 mmol TE 100 g−1 d.w.) was significantly different only in comparison to the WLDPE × OF and BLDPE × ChF variants, which registered values of 0.43 mmol TE 100 g^−1^ d.w.

The interaction effects among factors also significantly influenced the levels of chlorophyll a, chlorophyll b, lycopene, and β-carotene in the leaves of the Bologna perennial wall rocket (Table 6). The WLDPE × OF combination of factors promoted the highest synthesis of chlorophyll a (101.45 mg·100 g^−1^ d.w.), with values slightly—though not significantly—higher only compared to those associated with the BLDPE × ChF and BLDPE × NF variants. Regarding chlorophyll b, it was determined that the combinations BLDPE × NF, WLDPE × NF, and WLDPE × ChF induced significant differences in its synthesis by the plants, with the associated values being statistically different from one another. In contrast, for lycopene content, significant differences were found only between the values of WLDPE × OF and NM × ChF variants. The values detected for the remaining variants did not differ significantly from the highest or lowest values observed.

The statistical analysis of β-carotene content revealed significant differences between the experimental variants, with the highest value (6.35 ± 0.33 mg·100 g^−1^ d.w.) recorded for the BLDPE × NF variant, which was significantly different from all other factor combinations. Conversely, the lowest values (3.80 ± 0.07 mg·100 g^−1^ d.w. for WLDPE × ChF and 3.86 ± 0.19 mg·100 g^−1^ d.w. for NM × OF) were significantly lower than those observed in the other experimental variants.

### 2.3. Dimensionality Reduction and Exploratory Causal Statistical Analysis of Data

The results of the principal component analysis (PCA) applied to the entire dataset, illustrated in Figure 7, revealed that three of the eight independent principal component axes had eigenvalues greater than 1. The eigenvalues of PC1 and PC2 were 5.34 and 3.44, respectively, and these two principal components together accounted for 67.96% of the total variability, indicating that the majority of the information regarding the growth and quality traits of perennial wall–rocket under the influence of mulching and fertilization regimes was captured by these two components. PC1 was strongly associated with yield (0.370), number of leaves (0.383), β-carotene (0.370), and chlorophyll b (0.301), indicating that it represents a productivity-related axis, reflective of vegetative growth and nutritional quality. In contrast, its negative contributors—LAI (−0.352), CO_2_ assimilation rate (−0.336), TPC (−0.227), and lycopene (−0.259)—suggest a potential trade-off between yield-related traits and stress response or antioxidant properties under less favorable conditions. PC2 was defined by positive loadings for tannins (0.475), chlorophyll a (0.414), lycopene (0.360), and chlorophyll b (0.327), and negative contributions from DPPH (−0.371), ABTS (−0.243), and CO_2_ assimilation rate (−0.277), highlighting its relevance to antioxidant and pigment variability (Table 7).

Biologically, the PCA demonstrates that mulching and fertilization regimes distinctly influence both agronomic performance and biochemical composition. Variants such as WLDPE × ChF clustered near productivity traits, while WLDPE × OF was associated with enhanced photosynthetic and antioxidant parameters. Conversely, BLDPE × NF was linked to increased pigment levels (chlorophyll b and β-carotene), and NM × NF, WLDPE × NF, and NM × ChF aligned with high levels of phenolic compounds and antioxidant capacity. These findings underscore how specific agronomic practices can modulate growth, pigment synthesis, and bioactive compound accumulation in perennial wall–rocket.

To better understand the effects of fertilization and mulching practices, PCA was conducted along with Pearson correlation analysis on subsets of data grouped by the type of fertilization regime or mulch used. The findings from these analyses, illustrated in Figure 8 and Figure 9, indicated that changes in fertilizer type or mulching practices have distinct effects on the measured variables, as they are positioned in different quadrants. For example, in the nonfertilized variants, the absence of mulch increased the contents of tannins and antioxidant compounds, measured as ABTS and DPPH. In contrast, the plants grown with the BLDPE mulch presented a relatively high CO_2_ assimilation rate and chlorophyll a, chlorophyll b, lycopene, and β−carotene contents, whereas those grown with the WLDPE mulch presented increased TPC contents in the leaves. Under organic fertilization, the highest levels for most of the analyzed traits were associated with WLDPE mulch. Chemical fertilization combined with BLDPE increased the dry matter content; DPPH, chlorophyll a, chlorophyll b, and β-carotene contents; and, in combination with WLDPE, the yield, number of leaves per plant, leaf area, CO_2_ assimilation rate, ABTS content, tannin content, and lycopene content increased (Figure 8c). Overall, the PCA results revealed that perennial wall–rocket experienced the greatest oxidative stress under the nonfertilization regime, regardless of the mulching practice. Additionally, the nonfertilization regime combined with WLDPE and BLDPE mulching is detrimental to perennial wall–rocket yield.

The Pearson correlation diagrams revealed that the fertilization regime or the type of mulch also influenced the relationships among the variables (Figure 8 and Figure 9). For example, in the absence of fertilization, yield is strongly negatively correlated with traits related to plant physiology (CO_2_ assimilation rate, chlorophyll a, and chlorophyll b with r < − 0.87). In contrast, yield is strongly positively correlated with traits associated with oxidative stress (TPC, DPPH, ABTS, and tannins with r > 0.61). Furthermore, yield was highly negatively correlated with the number of leaves per nest and dry matter content, whereas the traits related to plant physiology were strongly negatively correlated with those associated with oxidative stress (Figure 8a). Organic fertilization generally induced moderate to strong positive correlations between morphological and physiological traits, whereas weakly negative to moderately positive correlations were found between morphophysiological and oxidative stress-related traits (Figure 8b).

## 3. Discussion

The present study demonstrates that the interactions between mulching practices and fertilization regimes significantly influence both the physiological and biochemical traits of the Bologna cultivar of perennial wall–rocket under northeastern Romanian climatic conditions. Notably, mulching with white polyethylene films significantly enhanced the CO_2_ assimilation rate compared to the unmulched version. However, despite this improvement in photosynthetic performance, no significant differences in agro-morphological traits were observed between mulched and non-mulched plants. This suggests that while physiological efficiency improved, it did not immediately translate into visible morphological changes across all treatments. Nevertheless, a significant distinction was observed between the two mulch types: plants grown under WLDPE exhibited superior agro-morphological characteristics compared to those under BLDPE. This indicates that white mulch may more effectively support vegetative development, possibly due to enhanced light reflectance and improved root zone temperature regulation. In contrast, BLDPE mulch had a notably positive effect on dry matter accumulation and β-carotene content, potentially due to its capacity to sustain higher soil temperatures, thereby indirectly promoting metabolic activity and carotenoid synthesis.

Furthermore, this study highlights the importance of harvesting period on yield, indicating a temporal component in optimizing production. Specifically, the findings demonstrated that BLDPE was the most effective at relatively cold ambient temperatures, resulting in the highest yield during the first harvest when the plants were grown at relatively low temperatures. In contrast, the yields from the other three harvests were lower than those from the WLDPE mulched or nonmulched variants, and these differences are directly related to the properties and advantages of the mulching films. Mulching with low-density polyethylene (LDPE) film is widely recognized for its ability to regulate soil temperature effectively [45,46]. According to Gheshm and Brown [47], black polyethylene mulch significantly increases soil temperature, whereas white-on-black polyethylene mulch maintains temperatures similar to those in bare soil plots. Tarara [46] reported that at a depth of 10 cm, soil temperatures were approximately 4 °C lower under white and aluminized reflective plastics than under black plastics and 1 to 2 °C lower than those in bare soil. The temperature reduction observed with the white film is attributed to its high reflectivity, which prevents excessive heat buildup. In addition, some of the reflected radiation reaches the lower leaves of plants, increasing illumination and photosynthesis, which can stimulate plant growth [46]. Thus, during cold periods, BLDPE enhances soil warming through its sunlight-attracting mechanism, creating a favorable microclimate for perennial wall–rocket growth. However, in hot conditions, it may cause overheating, reducing yield compared with white film, which reflects sunlight and maintains a more stable soil temperature. Song et al. [48] reported positive results for white mulch film in a study on garlic, whereas Zhao et al. [49] reported similar outcomes for corn. However, in melon cultivation—a crop that requires high temperatures—white mulch film is associated with the lowest yield [32]. Caruso’s study [50] reinforced the idea that perennial wall–rocket grows better with reduced sunlight exposure and hence temperature, demonstrating improved performance in summer when shaded.

The evaluation of the individual influence of the fertilization regime revealed that, overall, variation in fertilization type had no significant effect on most of the traits analyzed. Significant differences were observed only in the CO_2_ assimilation rate and DPPH antioxidant activity. Specifically, organic fertilization led to an increase in the CO_2_ assimilation rate and a decrease in DPPH antioxidant activity compared to the chemical and unfertilized regimes. Even if, from a statistical point of view, the differences found between fertilization regimes are statistically insignificant, the findings of this study evidenced that organic fertilization promotes better plant growth and development. Therefore, the lower yields observed at harvest from chemical fertilization, in comparison with organic and nonfertilized methods, may be attributed to lower temperatures that reduce nutrient availability in the soil [51]. Additionally, chemical fertilizers can negatively impact the activity of soil microorganisms, which are already affected by cold conditions [52]. In contrast, soils enriched with organic matter support a more resilient and active microbial community, even at lower temperatures, thereby enhancing nutrient mineralization and availability to plants [52]. The lowest yield at the fourth harvest under chemical fertilization may be due to the fact that chemical fertilizers at high temperatures can exacerbate plant stress, as the fast release of nutrients can overwhelm the ability of plants to absorb them properly. Instead, organic fertilizers release nutrients more gradually, which aligns better with the plant’s needs, thus reducing stress [44,53]. This may explain why organic farming systems, which emphasize soil health and microbial balance, often result in improved crop productivity [54]. Supporting these observations, Stanojković-Sebić [20] reported higher yields in arugula under organic fertilization compared to control and chemically fertilized treatments. In contrast, Teliban et al. [36] demonstrated that the highest yield of perennial wall–rocket, at 25.3 t·ha^−1^, was achieved in an experimental variant that used chemical fertilization along with white polyethylene film mulch. Additional positive responses to organic fertilization have been reported in quinoa and broccoli, as demonstrated by the studies of Chirita et al. [55] and Fracchiolla et al. [56].

Furthermore, the results of this study clearly demonstrate that the interaction between mulching practices and fertilization regimes significantly affects the physiological performance of perennial wall–rocket, particularly regarding CO_2_ assimilation rates. The chemical fertilization, especially in combination with BLDPE, resulted in a significant reduction of assimilation rate in plants. This decline may be attributed to the accumulation of salts from chemical fertilizers. Chemical fertilization is known to lead to the accumulation of salts in the soil, which creates osmotic stress in plants. High salt concentrations make it more difficult for plants to absorb water and nutrients, causing them to expend more energy to take up what they need. As a result, plants may become dehydrated, wilt, and struggle to grow properly. The increased energy demand also reduces the ability of plants to perform vital processes such as photosynthesis, further stunting growth and lowering yields. Additionally, excess salts can harm roots, limiting their function and contributing to nutrient imbalances [57]. Furthermore, the detrimental effects of chemical fertilization on plants are intensified by heat stress [58,59], as soil temperatures rise under black polyethylene mulch [46,47].

Regarding the functional compounds content in leaves was found that while the combinations of mulching and fertilization factors had no significant influence on total polyphenol content, significant changes in tannin levels and the antioxidant activity were observed. Specifically, the highest concentrations of tannins, TPC, and antioxidant activity were recorded in the NM × NF variant, which suggests that the plants in this group experienced the greatest oxidative stress. This increase in antioxidant compounds is likely a result of nutrient deficiency, a common stressor that triggers the production of phenolic compounds and antioxidants as a protective mechanism. The findings support the idea that a lack of fertilization, particularly under non-mulched conditions, leads to higher stress in perennial wall–rocket, stimulating the synthesis of these compounds as a defense mechanism [60,61,62]. Interestingly, no significant differences were observed in the polyphenol content across the various mulching and fertilization treatments. This may be explained by the fact that perennial wall–rocket is a species with a short vegetative cycle and low nutritional requirements, demonstrating high resilience to fluctuations in mulching and fertilization regimes. This resilience likely explains the absence of significant variation in the polyphenolic compound content, even though other bioactive compounds (e.g., tannins and antioxidants) exhibited more noticeable changes. In line with these observations, Hata et al. [63] reported that, compared with mineral fertilizer, unfertilized romaine lettuce contains higher concentrations of phenolic compounds (TPCs) and antioxidant compounds. In the control group, the TPC reached 1234.99 mg GAE 100 g^−1^ of d.w., whereas the DPPH level was 119.97 µmol TE g^−1^ of d.w. In contrast, the romaine lettuce that received mineral fertilization presented a TPC level of 960.22 mg GAE 100 g^−1^ of d.w. and a DPPH level of 91.95 µmol TE g^−1^ of d.w. So, this reinforces the idea that the absence of fertilization can promote an increase in antioxidant production, potentially as a response to nutrient stress.

The elevated tannin content in the NF × NM variant may suggest that these plants experienced water deficiency, as mulching is known to reduce soil water evaporation. Tannins are primary defense compounds that plants produce in response to abiotic stresses such as drought, heat, and high UV radiation, as well as pest attacks [64].

The interaction between mulching and fertilization also influences the levels of lipophilic pigments in perennial wall–rocket in different ways. The results revealed that the combination of white mulch and organic fertilization, which also resulted in high crop output, resulted in the highest concentrations of chlorophyll a. Although the chlorophyll a levels under WLDPE × OF were not statistically different from those under BLDPE × ChF and BLDPE × NF, the results underscore the potential of combining reflective mulch with organic inputs to enhance photosynthetic efficiency through improved pigment biosynthesis. In the case of chlorophyll b, a more nuanced interaction was observed. Significant differences were recorded among BLDPE × NF, WLDPE × NF, and WLDPE × ChF, indicating that both mulch color and fertilization type play roles in modulating chlorophyll composition. These differences suggest that the physiological responses of wall–rocket may be closely tied to how light reflection and soil nutrient availability interact to influence pigment biosynthesis. In terms of lycopene accumulation, statistically significant differences were identified only between the WLDPE × OF and NM × ChF treatments. The relatively limited variation among other treatments implies that lycopene content is either less responsive to changes in cultivation inputs or is governed by a more complex interaction of environmental and genetic factors.

The analysis of β-carotene content revealed that the interaction between mulching practices and fertilization regimes led to a higher degree of significant variation among the experimental variants, with the BLDPE × NF combination resulting in the highest concentration, a value that significantly exceeded all other experimental variants. This finding likely reflects a synergistic effect between the warming properties of black mulch and the absence of fertilization-induced nutrient imbalances, conditions that may favor the biosynthesis of carotenoids. According to Tang et al. [14], mulching enhances the chlorophyll content in plants primarily by improving soil moisture, regulating temperature, and increasing nutrient availability. The positive influence of mulching films on pigment content was also reported by de Shah Jahan et al. [65] in salad. Although fertilization type individually did not cause significant differences across all pigment levels, chemical fertilization consistently resulted in the lowest pigment values. Conversely, organic fertilization supported the highest concentrations of chlorophyll a and lycopene. This trend aligns with findings by Ikyo et al. [66], who reported lower pigment content in *Amaranthus* spp. and *Corchorus olitorius* leaves under chemical fertilization compared to organic fertilization with pig manure. The increase in pigment content under organic fertilization is attributed primarily to the gradual and sustained release of mineral nutrients, which support chlorophyll synthesis in plants throughout the growing period [67].

PCA and Pearson analyses were used in this study to identify the key patterns and relationships in the growth and quality traits of perennial wall–rocket, since their usefulness in the statistical analysis of the data has been demonstrated in various studies [68,69,70,71]. The PCA results underscored the substantial influence that combined agronomic practices—specifically mulching and fertilization—have on the plant’s biochemical and physiological profiles. Notably, the combination of organic fertilization and WLDPE mulching was associated with enhanced morphological and physiological performance. This combination was positioned at a favorable location in the PCA biplot, aligning closely with traits such as photosynthetic rate, chlorophyll a content, and yield, suggesting it promotes optimal plant development under the tested conditions. The results from the Pearson correlation analysis revealed that both organic fertilization and WLDPE film resulted in weakly negative and moderately strong positive correlations between morphological and physiological traits, whereas the other fertilization regimes and mulching practices caused a negative correlation between the traits. For example, regardless of the fertilization regime, there are moderate to strong positive correlations between yield and leaf water content, indicating that adequate leaf water availability is maintained. However, when BLDPE mulching film was used, yield and leaf water content showed a strong negative correlation. These findings suggest that the mulching film may induce stress, potentially leading to excessively high soil temperatures or altered root-zone aeration, which negatively impacts the relationship between leaf water content and yield. In terms of mulching practices, positive correlations between morphological and physiological characteristics were identified when WLDPE mulching film was used. Furthermore, negative correlations were detected between morpho-physiological and oxidative stress traits. This implies that WLDPE mulching film supports plant growth by promoting favorable interactions between morphological and physiological traits. Therefore, organic fertilization combined with WLDPE film mulching effectively supports the growth of perennial wall–rocket under the specific climatic conditions of the study area, optimizing both the structural development and physiological functions of the plant.

## 4. Materials and Methods

### 4.1. Design of the Experiment and Research Protocol

The experiment was carried out in 2022 and 2023 during the spring–summer crop cycle in a plastic tunnel with ventilation systems at the top of the entrances. It was located on the premises of the ‘V. Adamachi’ Farm, part of the Iasi University of Life Sciences. In the autumn off-season, barley was grown in the tunnel and later incorporated into the soil to facilitate the restoration of soil structure and achieve uniformity in mineral content. At the time of incorporation, the barley plants were approximately 15 cm tall.

The effects of white polyethylene film (WLDPE), black polyethylene film (BLDPE), and nonmulched (NM) practices, in combination with organic fertilization (OF), chemical fertilization (ChF), and nonfertilization (NF) regimes, on ‘Bologna’ perennial wall–rocket were investigated.

Crop fertilization was performed in 4 stages. The initial stage, known as base fertilization, was performed prior to planting, while the subsequent three stages were carried out after each harvest. For organic fertilization, chicken manure with the following composition was used: organic nitrogen, 4%; phosphorus, 4%; water-soluble potassium, 4%; and organic matter, 72%. Chemical fertilization consisted of the application of soluble NPK, which contained 23% total nitrogen (17% ammonia nitrogen and 6% nitrate nitrogen), 5% phosphoric anhydride, 5% potassium oxide, 29% sulfuric anhydride, 0.10% iron, 0.05% manganese, and 0.10% zinc. A total of 28 kg NPK for the chemically fertilized variant and 200 kg for the organically fertilized version were applied per 1000 m^2^. Before application, the fertilizer was dissolved in water. Following the main fertilization, the soil was mulched using a 60 μm-thick black or white polyethylene film, as illustrated in Figure 10. The fertilizer doses were determined in accordance with the provisions of EU Regulation 848/2018 on organic production, which permits the application of up to a maximum of 170 kg N/ha/year [72]. Furthermore, considering that only 60% of the nitrogen, phosphorus, and potassium from organic fertilizer is available for plant uptake in the year of application [73] and that two rocket crops can be cultivated on the same plot within a single year, the fertilizer application rates were calculated accordingly.

Experiments were conducted in triplicate, with each replicate consisting of 12 nests. For each nest in an alveolar tray, 20 seeds and the seedlings were grown in a greenhouse at 18–20 °C day/16–18 °C night and 70–75% relative humidity (RH). The plants were planted in a polytunnel 30 days after germination. In both years, sowing took place on 6 March, and planting was conducted on 7 April. The first harvest occurred 30 days after planting in the field, followed by subsequent harvests (second, third, and fourth) at 30-day intervals. During warm days, the temperature in the tunnel was regulated by opening the arches to ensure adequate ventilation. Additionally, a layer of calcium carbonate (lime) was applied on the tunnel surface [74].

The variations in temperature and atmospheric humidity at the plant level during the experimental period were monitored and recorded via an electronic Data Logger^®^ device. The data registered were plotted in Figure 11.

The perennial wall–rocket leaves were harvested early in the morning, at 3–5 cm above the cotyledons, to avoid damaging the regenerating shoots [75]. The samples were immediately sent to the laboratory and stored in a refrigerator to maintain their turgor until analysis, which occurred on the same day. All maintenance work during the growing season was performed according to the literature [6,76].

### 4.2. Analytical Methods for the Evaluation of Analyzed Parameters

The results of the quantitative analyses were summarized by items such as mass, number of leaves, leaf area per nest, and yield. Quality analyses refer to the percentage of dry matter and water, photosynthetic assimilation rate (A), antioxidant activity (DPPH and ABTS), total phenolic content (TPC), tannins, chlorophyll a, chlorophyll b, lycopene, and β-carotene.

Leaf gas exchange parameters were analyzed via a portable LCpro T^®^ (ADC BioScientific Ltd., Hoddesdon, Hertfordshire, UK) intelligent photosynthesis system [37].

The leaf area index (LAI) was analyzed via a Li-3100^®^ surface area meter manufactured by LICOR, Inc., Lincoln, NE, USA. The dry mass of each sample was determined by drying in an MOV-112F oven (SANYO Electric Co., Ltd., Osaka, Japan) at 70 °C until a constant weight was achieved [29]. The dried samples were subsequently used to determine the contents of functional compounds and lipophilic pigments.

The extraction of functional compounds from perennial wall–rocket samples consisted of mixing 2 mg of dry sample with 1 mL of 80% methanol and 20% formic acid solution (phase A). The mixture was subjected to ultrasonication for 10 min, followed by centrifugation at 15,000 rpm and 4 °C for 15 min. Afterward, the liquid phase was collected, and the biomass pellet was again subjected to the extraction procedure described above. The liquid phase collected from the two extractions was brought up to 2 mL with phase A and stored at −80 °C until the analysis of total phenolics, tannins, and antioxidant activity [77].

The total polyphenol content (TPC) was determined via the Folin–Ciocalteu reagent following the Slinkard–Singleton method [78]. The procedure consisted of mixing 10 µL of hydrophilic extract with 175 µL of distilled water and 12 µL of Folin–Ciolcalteu reagent. After allowing the mixture to react for 3 min, 30 µL of a 20% aqueous sodium carbonate solution was added, and the samples were kept in the dark for 60 min. After this incubation period, their absorbance was measured at 765 nm via a microplate spectrophotometer (Thermo Scientific Multiskan GO, Vantaa, Finland). The data obtained were compared with a series of known concentrations of gallic acid standards, which were prepared using the same method. The results were expressed as gallic acid equivalents (GAE) in milligrams per 100 g dry weight (mg GAE·100 g^−1^ D.W.).

The tannin content was determined via the vanillin method. The procedure consisted of adding 25 µL of sample, 150 µL of vanillin solution, and 75 µL of HCl to a reaction tube. The mixture was homogenized and incubated for 20 min at room temperature. After incubation, the absorbance of the solution was measured at 500 nm via a spectrophotometer. The concentration of tannins was determined by comparing the sample’s absorbance to a standard curve established with catechin [77].

The antioxidant activity was measured via the ABTS test at 730 nm via a Thermo Scientific Multiskan GO microplate spectrophotometer. The samples were prepared by mixing 175 µL of distilled water, 190 µL of ABTS solution, and 10 µL of plant extract and then keeping them in the dark for 15 min before absorbance measurement. The ABTS solution was prepared by dissolving 38.6 mg of ABTS reagent in 10 mL of 2.45 mM potassium persulfate solution (334.4 mg dissolved in 500 mL of distilled water). The solution was incubated for 16 h in the dark to allow the formation of the ABTS radical before use in the analysis. Trolox standards were prepared in 0.1 mM methanol, and the calibration curve was obtained following the same steps as those for the sample [77].

Antioxidant activity was determined via the DPPH test. A 3.5 mg sample of the DPPH radical was dissolved in 10 mL of methanol and adjusted to an absorbance of 0.98 at 515 nm. In each microplate well, 100 µL of DPPH solution and 10 µL of either Trolox or the sample were added. The initial absorbance was recorded at 515 nm, followed by the addition of methanol or DPPH radical, on the basis of the obtained readings. After the reaction period, the absorbance was measured again at 515 nm via a microplate spectrophotometer (Thermo Scientific Multiskan GO), with the decrease in absorbance indicating the antioxidant activity of the sample [77].

The extraction of lipophilic pigments was conducted under low-light conditions to prevent pigment degradation, following the isolation protocol described by Nagata and Yamashita [79]. The extraction procedure involved mixing 0.2 g of dry material with 1 mL of a hexane and acetone mixture (4:6, *v*:*v*), followed by centrifugation at 15,000 rpm for 15 min. The liquid phase was collected, and the pellet was subjected to a second extraction. Both liquid phases were subsequently stored at −80 °C until analysis. The absorbance of the extract was measured at 453, 505, 645, and 663 nm via a Synergy HTX multimode microplate reader. The concentrations of chlorophyll a, chlorophyll b, β-carotene, and lycopene were calculated via the formulas provided by Nagata and Yamashita [79].

### 4.3. Statistical Analysis of the Results

Multiple statistical techniques were used to analyze the relationships between variables and to extract the relevant information from the dataset obtained in this study. Initially, ANOVA was performed to determine the presence of statistically significant differences in the data. This was followed by a means comparison conducted with Duncan’s test at a significance level of *p* ≤ 0.05. This part of the statistical analysis was performed in SPSS version 26, and the results are reported as the means ± standard deviations. Additionally, principal component analysis (PCA) was employed to reduce the dimensionality of the data while retaining important variability, effectively summarizing the dominant patterns and relationships. Pearson correlation coefficients were calculated to evaluate the linear relationships among the analyzed traits. Both analyses were conducted via the OriginLab Pro 2025 free trial (OriginLab Corporation, Northampton, MA, USA). Together, these methods provide insights into how fertilization regimes and mulching practices impact perennial wall–rocket, highlighting the significant changes in traits and the relationships between variables.

## 5. Conclusions

The findings of this study underscore the critical role that mulching and fertilization play in patterning the physiological and biochemical responses of perennial wall–rocket *(Diplotaxis tenuifolia* cv. Bologna) under northeastern Romanian climatic conditions. The interaction between these factors not only influenced the growth parameters, such as yield and leaf area, but also impacted the synthesis of key compounds such as chlorophylls, β-carotene, and antioxidants.

The fertilization regime alone had no significant effect on most parameters, with the exception of CO_2_ assimilation and DPPH antioxidant activity, which were positively and negatively affected, respectively, by organic fertilization.

Although the maximum yields per harvest varied, mulching with white film, in conjunction with organic or chemical fertilization, generally supported strong yields. The CO_2_ assimilation rate was significantly enhanced under WLDPE × OF (15.07 µmol m^−2^ s^−1^), while combinations such as BLDPE × ChF and NM × NF recorded the lowest values, indicating negative effects of chemical fertilization and absence of mulching under certain conditions. Dry matter content was highest in the BLDPE × ChF variant (9.27%), with statistically significant differences compared to WLDPE × ChF (8.05%). Chlorophyll a content peaked at 101.45 mg·100 g^−1^ d.w. under WLDPE × OF, while β-carotene was highest in the BLDPE × NF variant (6.35 mg·100 g^−1^ d.w.), significantly surpassing all other combinations. Tannin content was significantly higher in the NM × NF variant, reflecting stress-induced antioxidant accumulation.

Overall, this study highlights the importance of combining mulching with appropriate fertilization practices to maximize both the agronomic and nutritional potential of perennial wall–rocket, providing insights into strategies that could enhance crop production and quality under diverse environmental conditions. Future research should explore the long-term effects of these treatments and their potential in optimizing both crop yield and nutritional content in commercial-scale production.

## Figures and Tables

**Figure 1 plants-14-01421-f001:**
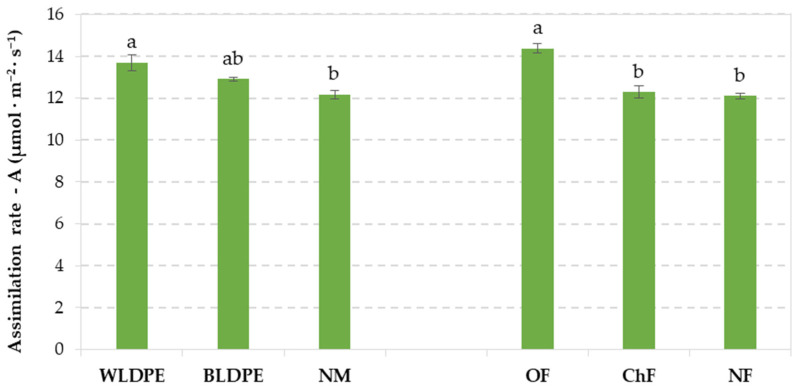
CO_2_ assimilation rate in Bologna perennial wall–rocket leaves under the individual influence of mulching and fertilization regimes. The values are presented as the means ± standard errors from three independent replications. Different lowercase letters indicate significant differences between groups at *p* ≤ 0.05 according to Duncan’s test, with ‘a’ representing the highest value. WLDPE—white polyethylene film; BLDPE—black polyethylene film; NM—nonmulched; OF—organic fertilization; ChF—chemical fertilization; NF—nonfertilized.

**Figure 2 plants-14-01421-f002:**
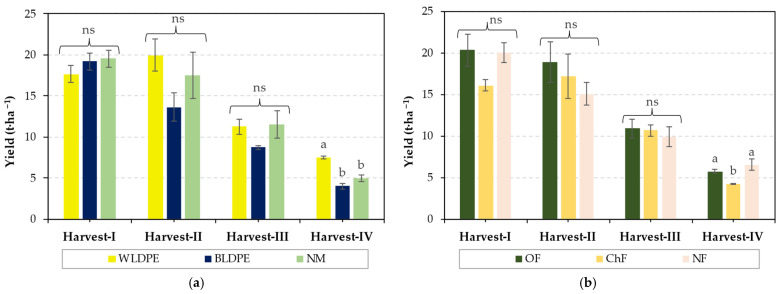
Harvest dynamics under the individual influence of (**a**) mulching and (**b**) fertilization regimes. The values are presented as the means ± standard errors from three independent replications. ns denotes nonsignificant differences, while different lowercase letters (a,b) at the same harvest indicate significant differences between groups at *p* ≤ 0.05 according to Duncan’s test, with ‘a’ representing the highest value. WLDPE—white polyethylene film; BLDPE—black polyethylene film; NM—nonmulched; OF—organic fertilization; ChF—chemical fertilization; NF—nonfertilized.

**Figure 3 plants-14-01421-f003:**
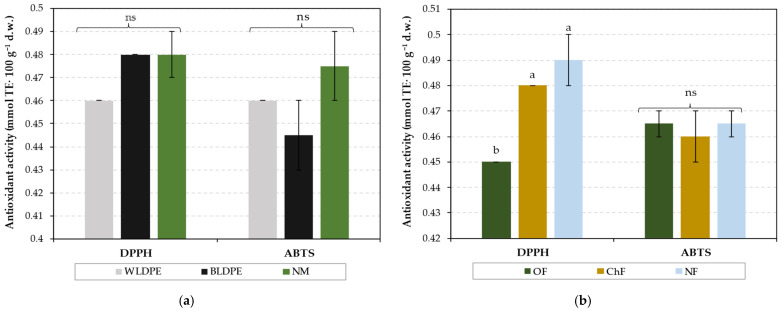
Individual effects of (**a**) mulching and (**b**) fertilization treatments on antioxidant activity in Bologna perennial wall–rocket leaves. The values are presented as the means ± standard errors from three independent replications. ns denotes nonsignificant differences, while different lowercase letters within the same antioxidant activity parameter indicate significant differences between groups at *p* ≤ 0.05 according to Duncan’s test, with ‘a’ representing the highest value. WLDPE—white polyethylene film; BLDPE—black polyethylene film; NM—nonmulched; OF—organic fertilization; ChF—chemical fertilization; NF—nonfertilized.

**Figure 4 plants-14-01421-f004:**
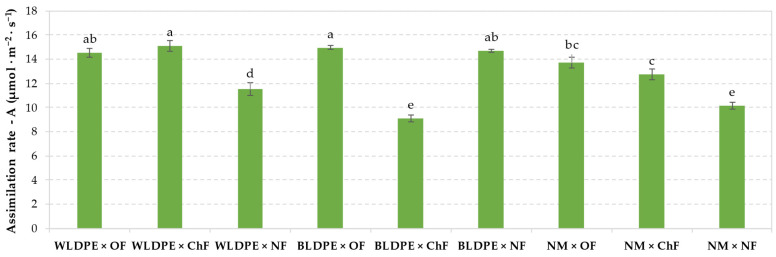
Effects of factor interactions on the CO_2_ assimilation rate in Bologna perennial wall–rocket leaves. The values are presented as the means ± standard errors from three independent replications. Different lowercase letters indicate significant differences between groups at *p* ≤ 0.05 according to Duncan’s test, with ‘a’ representing the highest value. WLDPE—white polyethylene film; BLDPE—black polyethylene film; NM—nonmulched; OF—organic fertilization; ChF—chemical fertilization; NF—nonfertilized.

**Figure 5 plants-14-01421-f005:**
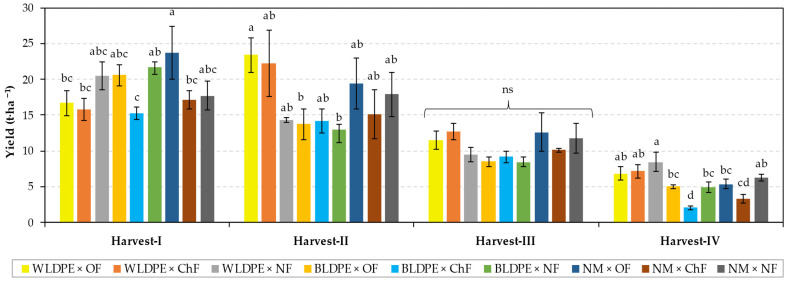
Influence of factor interactions on harvest dynamics of Bologna perennial wall–rocket. The values are presented as the means ± standard errors from three independent replications. ns denotes nonsignificant differences, while different lowercase letters at the same harvest indicate significant differences between groups at *p* ≤ 0.05 according to Duncan’s test, with ‘a’ representing the highest value. WLDPE—white polyethylene film; BLDPE—black polyethylene film; NM—nonmulched; OF—organic fertilization; ChF—chemical fertilization; NF—nonfertilized.

**Figure 6 plants-14-01421-f006:**
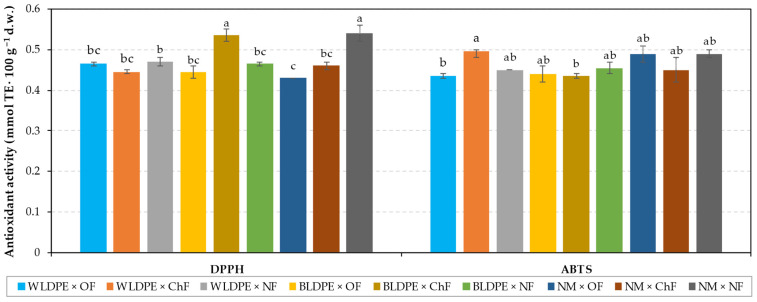
Effects of factor interactions on antioxidant activity in Bologna perennial wall–rocket leaves. The values are presented as the means ± standard errors from three independent replications. Different lowercase letters within the same antioxidant activity parameter indicate significant differences between groups at *p* ≤ 0.05 according to Duncan’s test, with ‘a’ representing the highest value. WLDPE—white polyethylene film; BLDPE—black polyethylene film; NM—nonmulched; OF—organic fertilization; ChF—chemical fertilization; NF—nonfertilized.

**Figure 7 plants-14-01421-f007:**
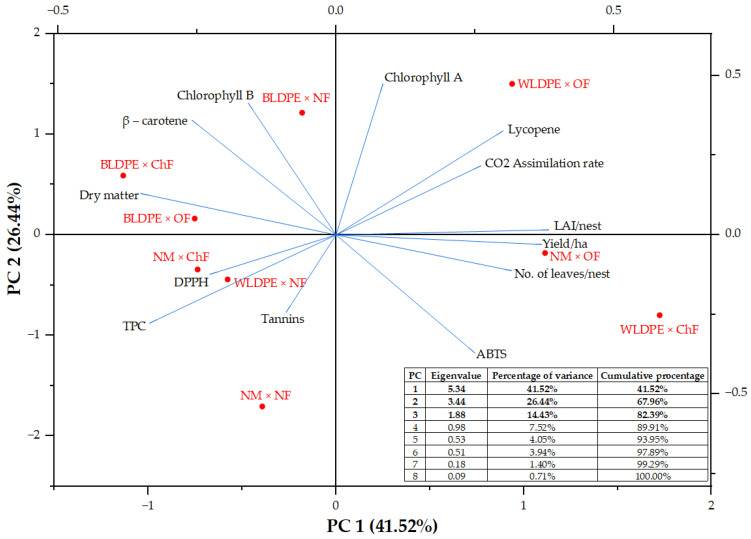
Principal component analysis score plot showing the variation in growth and quality traits of perennial wall–rocket under different mulching and fertilization treatments.

**Figure 8 plants-14-01421-f008:**
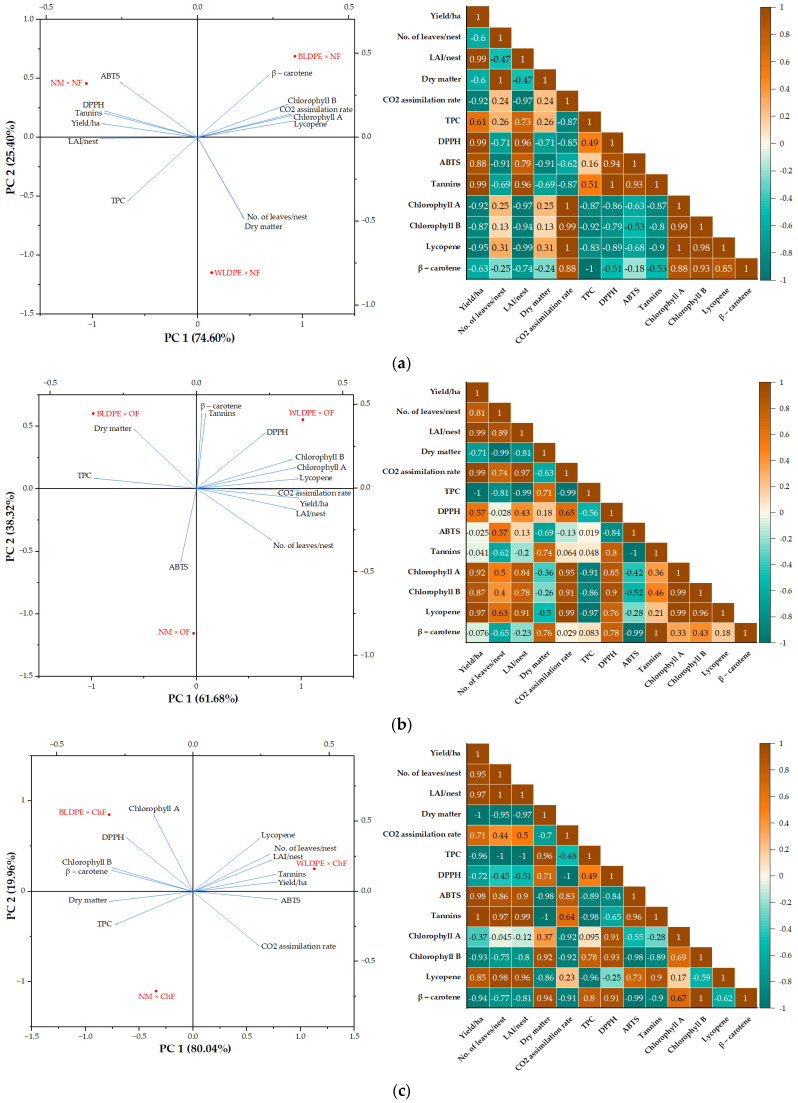
PCA score plot and Pearson correlation diagram showing the variation in growth and quality traits of perennial wall–rocket under the influence of (**a**) nonfertilized, (**b**) organic fertilization, and (**c**) chemical fertilization regimes.

**Figure 9 plants-14-01421-f009:**
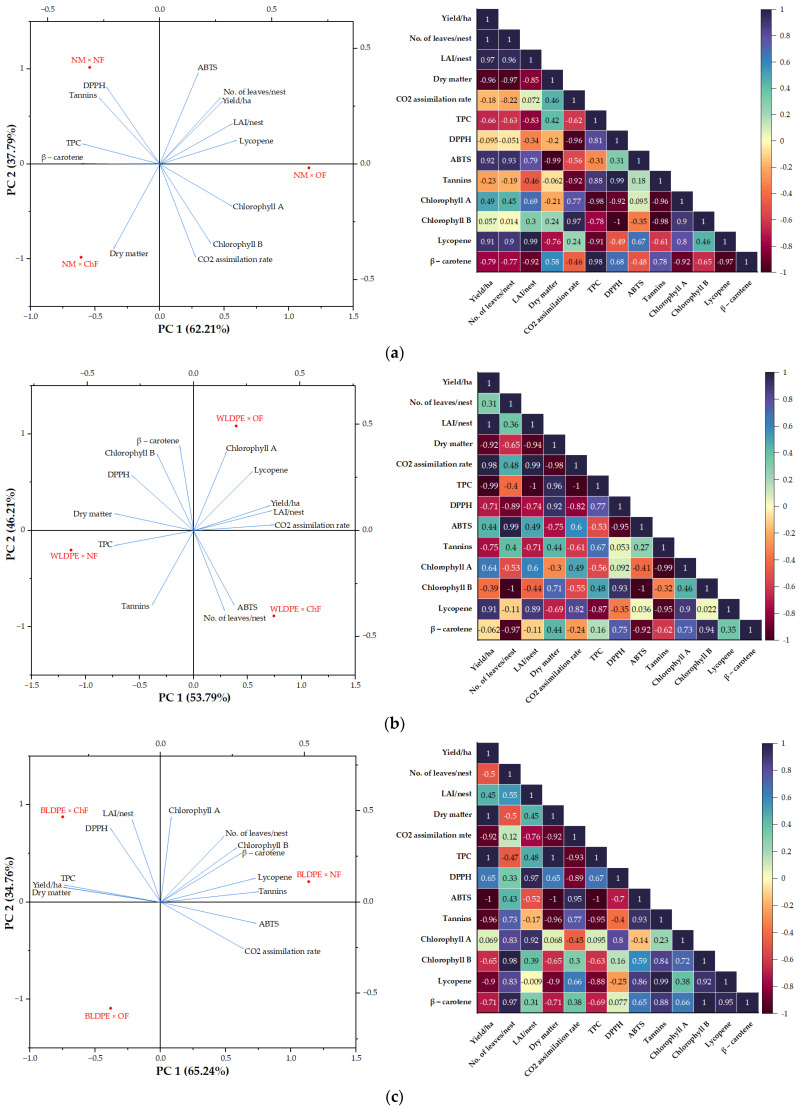
PCA score plot and Pearson correlation diagram showing the variation in growth and quality traits of perennial wall–rocket under the influence of (**a**) nonmulching, (**b**) WLDPE, and (**c**) BLDPE mulch.

**Figure 10 plants-14-01421-f010:**
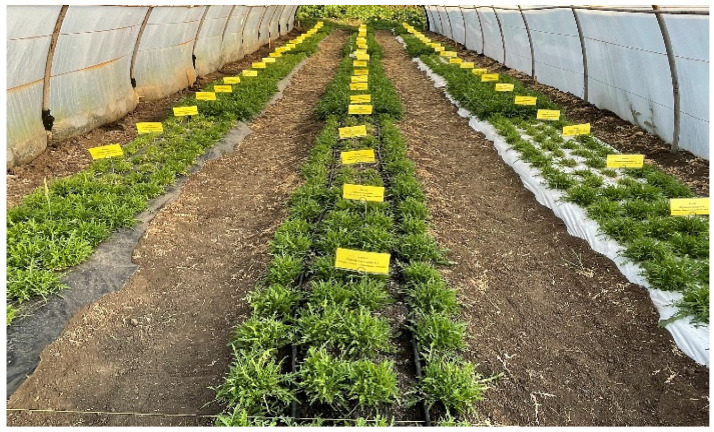
Experimental protocol scheme (original).

**Figure 11 plants-14-01421-f011:**
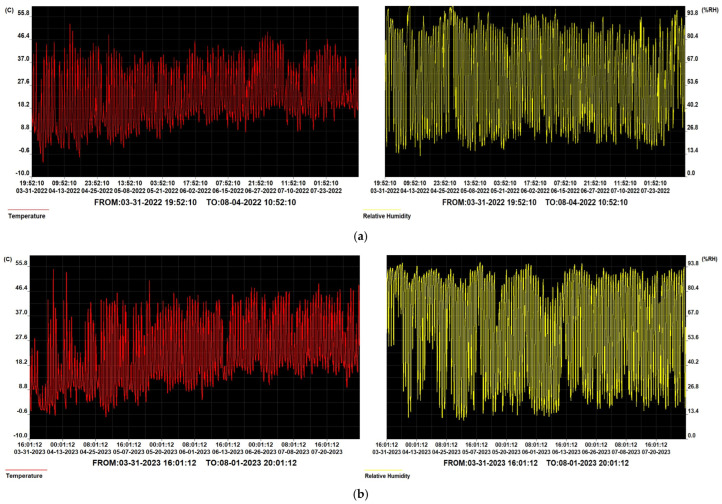
Trends of the temperature and relative humidity inside the polytunnel during the experiments in (**a**) 2022 and (**b**) 2023.

**Table 1 plants-14-01421-t001:** Individual influences of the studied factors on the agro–morphological traits of the Bologna perennial wall–rocket.

Experimental Variant	Leaves/Nest	LAI (cm^2^·cm^−2^)	Total Yield	
			Nest (g)	t·ha^−1^
WLDPE	660.67 ± 24.20	11,309.75 ± 416.52 a	633.31 ± 25.22 a	56.36 ± 2.24 a
BLDPE	577.33 ± 23.30	9481.05 ± 269.29 b	511.64 ± 11.76 b	45.54 ± 1.05 b
NM	634.00 ± 27.05	10,895.45 ± 505.19 ab	601.38 ± 42.59 ab	53.52 ± 3.79 ab
Significance	ns	*	*	*
OF	636.44 ± 17.77	11,027.83 ± 330.97	627.24 ± 18.42	55.82 ± 1.64
ChF	591.67 ± 13.83	10,090.26 ± 562.68	540.61 ± 39.11	48.11 ± 3.48
NF	643.89 ± 39.86	10,568.16 ± 422.30	578.47 ± 19.98	51.48 ± 1.78
Significance	ns	ns	ns	ns

The values are presented as the means ± standard errors from three independent replications. Different lowercase letters in the same column indicate significant differences between groups at *p* ≤ 0.05 according to Duncan’s test, with ‘a’ representing the highest value. * indicates significant differences between the compared means, whereas ns denotes nonsignificant differences. WLDPE—white polyethylene film; BLDPE—black polyethylene film; NM—nonmulched; OF—organic fertilization; ChF—chemical fertilization; NF—nonfertilized.

**Table 2 plants-14-01421-t002:** Individual influences of factors on water content, dry weight, total phenolic content and tannin contents in Bologna perennial wall–rocket leaves.

Experimental Variant	Dry Matter (%)	TPC (mg GAE·100 g^−1^ d.w.)	Tannins (mmol·100 g^−1^ d.w.)
WLDPE	8.29 ± 0.07 b	1.95 ± 0.11	0.06 ± 0.01
BLDPE	8.95 ± 0.09 a	1.98 ± 0.15	0.06 ± 0.01
NM	8.59 ± 0.11 b	2.04 ± 0.08	0.06 ± 0.01
Significance	*	ns	ns
OF	8.51 ± 0.07	1.91 ± 0.12	0.06 ± 0.01
ChF	8.79 ± 0.08	1.99 ± 0.13	0.06 ± 0.01
NF	8.53 ± 0.12	2.07 ± 0.10	0.06 ± 0.00
Significance	ns	ns	ns

The values are presented as the means ± standard errors from three independent replications. Different lowercase letters in the same column indicate significant differences between groups at *p* ≤ 0.05 according to Duncan’s test, with ‘a’ representing the highest value. * Indicates significant differences between the compared means, whereas ns denotes nonsignificant differences. WLDPE—white polyethylene film; BLDPE—black polyethylene film; NM—nonmulched; OF—organic fertilization; ChF—chemical fertilization; NF—nonfertilized.

**Table 3 plants-14-01421-t003:** Individual influences of the studied factors on the leaf pigment content of the Bologna perennial wall–rocket.

Experimental Variant	Chlorophyll a (mg·100 g^−1^ d.w.)	Chlorophyll b (mg·100 g^−1^ d.w.)	Lycopene (mg·100 g^−1^ d.w.)	β-Carotene (mg·100 g^−1^ d.w.)
WLDPE	93.14 ± 1.97	36.00 ± 0.33	9.76 ± 0.21	4.77 ± 0.05 b
BLDPE	93.45 ± 0.82	39.12 ± 0.83	9.40 ± 0.19	5.99 ± 0.23 a
NM	87.24 ± 2.86	35.70 ± 1.18	9.17 ± 0.27	4.6 ± 0.26 b
Significance	ns	ns	ns	*
OF	94.92 ± 0.15	37.05 ± 0.18	9.73 ± 0.05	5.03 ± 0.29
ChF	91.21 ± 2.25	35.81 ± 1.04	9.29 ± 0.19	4.91 ± 0.20
NF	89.25 ± 1.17	38.18 ± 1.02	9.41 ± 0.22	5.36 ± 0.01
Significance	ns	ns	ns	ns

The values are presented as the means ± standard errors from three independent replications. Different lowercase letters in the same column indicate significant differences between groups at *p* ≤ 0.05 according to Duncan’s test, with ‘a’ representing the highest value. * Indicates significant differences between the compared means, whereas ns denotes nonsignificant differences. WLDPE—white polyethylene film; BLDPE—black polyethylene film; NM—nonmulched; OF—organic fertilization; ChF—chemical fertilization; NF—nonfertilized.

**Table 4 plants-14-01421-t004:** Effect of factor interactions on the agromorphological traits of the Bologna perennial wall–rocket.

Experimental Variant	Leaves/Nest	LAI (cm^2^·cm^−2^)	TOTAL YIELD	
			Nest (g)	t·ha^−1^
WLDPE × OF	655.00 ± 27.75 a	11,328.77 ± 196.48 ab	656.78 ± 10.13 ab	58.45 ± 0.90 ab
WLDPE × ChF	670.67 ± 36.55 a	11,556.13 ± 1245.31 ab	650.73 ± 83.00 abc	57.91 ± 7.39 abc
WLDPE × NF	656.33 ± 28.85 a	11,044.36 ± 265.19 ab	592.42 ± 9.34 abcd	52.73 ± 0.83 abcd
BLDPE × OF	591.00 ± 20.07 ab	9715.37 ± 443.82 bc	537.68 ± 23.29 bcd	47.85 ± 2.07 bcd
BLDPE × ChF	519.67 ± 35.03 b	8842.67 ± 450.49 c	457.10 ± 22.34 d	40.68 ± 1.99 d
BLDPE × NF	621.33 ± 29.49 ab	9885.11 ± 513.22 abc	540.12 ± 22.92 bcd	48.07 ± 2.04 bcd
NM × OF	663.33 ± 31.17 a	12,039.34 ± 970.78 a	687.26 ± 66.15 a	61.17 ± 5.89 a
NM × ChF	584.67 ± 15.07 ab	9871.99 ± 325.64 abc	514.01 ± 28.82 cd	45.75 ± 2.56 cd
NM × NF	654.00 ± 66.30 a	10,775.01 ± 681.02 abc	602.87 ± 41.90 abc	53.66 ± 3.73 abc

The values are presented as the means ± standard errors from three independent replications. Different lowercase letters in the same column indicate significant differences between groups at *p* ≤ 0.05 according to Duncan’s test, with ‘a’ representing the highest value. WLDPE—white polyethylene film; BLDPE—black polyethylene film; NM—nonmulched; OF—organic fertilization; ChF—chemical fertilization; NF—nonfertilized.

**Table 5 plants-14-01421-t005:** Effects of interaction factors on water content, dry weight, total phenolic content, and tannin contents in Bologna perennial wall–rocket leaves.

Experimental Variant	Dry Matter (%)	TPC (mg GAE·100 g^−1^ d.w.)	Tannins (mmol·100 g^−1^ d.w.)
WLDPE × OF	8.21 ± 0.01 cd	1.84 ± 0.12	0.06 ± 0.01 abc
WLDPE × ChF	8.05 ± 0.10 d	1.84 ± 0.02	0.06 ± 0.01 abc
WLDPE × NF	8.61 ± 0.29 bc	2.17 ± 0.19	0.06 ± 0.00 ab
BLDPE × OF	8.91 ± 0.20 ab	1.99 ± 0.17	0.06 ± 0.01 abc
BLDPE × ChF	9.27 ± 0.08 a	2.05 ± 0.17	0.05 ± 0.00 bc
BLDPE × NF	8.65 ± 0.03 bc	1.90 ± 0.12	0.06 ± 0.00 ab
NM × OF	8.40 ± 0.01 cd	1.90 ± 0.04	0.05 ± 0.01 c
NM × ChF	9.04 ± 0.24 ab	2.10 ± 0.20	0.06 ± 0.01 abc
NM × NF	8.33 ± 0.10 cd	2.14 ± 0.01	0.07 ± 0.01 a
Significance	*	ns	*

The values are presented as the means ± standard errors from three independent replications. Different lowercase letters in the same column indicate significant differences between groups at *p* ≤ 0.05 according to Duncan’s test, with ‘a’ representing the highest value. * Indicates significant differences between the compared means, whereas ns denotes nonsignificant differences. WLDPE—white polyethylene film; BLDPE—black polyethylene film; NM—nonmulched; OF—organic fertilization; ChF—chemical fertilization; NF—nonfertilized.

**Table 6 plants-14-01421-t006:** Effects of interaction factors on chlorophyll a, chlorophyll b, lycopene, and β-carotene contents in Bologna perennial wall–rocket leaves.

Experimental Variant	Chlorophyll a (mg·100 g^−1^ d.w.)	Chlorophyll b (mg·100 g^−1^ d.w.)	Lycopene (mg·100 g^−1^ d.w.)	β-Carotene (mg·100 g^−1^ d.w.)
WLDPE × OF	101.45 ± 3.74 a	38.47 ± 0.90 ab	10.16 ± 0.36 a	5.74 ± 0.31 abc
WLDPE × ChF	89.51 ± 1.15 bcd	32.47 ± 0.08 c	9.73 ± 0.16 ab	3.80 ± 0.07 d
WLDPE × NF	88.45 ± 1.00 bcd	37.07 ± 0.01 b	9.38 ± 0.10 ab	4.77 ± 0.09 cd
BLDPE × OF	87.79 ± 2.33 bcd	35.87 ± 0.29 bc	9.19 ± 0.07 ab	5.68 ± 0.55 abc
BLDPE × ChF	97.45 ± 5.17 ab	39.05 ± 2.16 ab	9.25 ± 0.51 ab	5.96 ± 0.19 ab
BLDPE × NF	95.13 ± 0.38 abc	42.45 ± 0.62 a	9.78 ± 0.14 ab	6.35 ± 0.33 a
NM × OF	90.88 ± 0.95 bcd	36.16 ± 0.02 bc	9.54 ± 0.01 ab	3.86 ± 0.19 d
NM × ChF	86.69 ± 2.73 cd	35.91 ± 1.06 bc	8.89 ± 0.23 b	4.98 ± 0.34 bc
NM × NF	84.16 ± 4.89 d	35.03 ± 2.45 bc	9.07 ± 0.61 ab	4.95 ± 0.25 bc

The values are presented as the means ± standard errors from three independent replications. Different lowercase letters in the same column indicate significant differences between groups at *p* ≤ 0.05 according to Duncan’s test, with ‘a’ representing the highest value. WLDPE—white polyethylene film; BLDPE—black polyethylene film; NM—nonmulched; OF—organic fertilization; ChF—chemical fertilization; NF—nonfertilized.

**Table 7 plants-14-01421-t007:** Eigenvalues of the correlation matrix showing the affinities of the growth and quality traits of perennial wall–rocket to PCs.

Traits	Extracted Eigenvectors							
	PC1	PC2	PC3	PC4	PC5	PC6	PC7	PC8
Yield/ha	0.370	−0.030	0.179	−0.334	0.375	−0.061	0.225	0.181
No. of leaves/nest	0.383	0.016	0.211	−0.341	0.078	0.014	0.141	0.185
LAI/nest	−0.352	0.130	−0.163	−0.310	0.013	0.460	−0.297	0.261
Dry matter	0.261	0.217	−0.389	0.321	0.287	0.132	0.003	0.603
CO_2_ assimilation rate	−0.336	−0.277	0.048	−0.075	−0.104	0.235	0.670	0.163
TPC	−0.227	−0.124	0.561	−0.182	−0.053	−0.227	−0.227	0.331
DPPH	0.250	−0.371	0.002	0.269	−0.220	−0.405	−0.177	0.203
ABTS	−0.090	−0.243	0.451	0.456	0.470	0.319	−0.010	0.044
Tannins	0.084	0.475	0.257	−0.224	−0.141	0.011	−0.111	0.130
Chlorophyll a	−0.158	0.414	0.157	0.327	−0.369	−0.185	0.380	0.320
Chlorophyll b	0.301	0.327	0.209	0.135	−0.006	0.194	0.225	−0.433
Lycopene	−0.259	0.360	0.197	0.221	0.303	−0.157	−0.205	−0.097
β-carotene	0.370	−0.030	0.179	−0.334	0.375	−0.061	0.225	0.181

## Data Availability

Data are contained within the article.

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
