# Peer review of "Effects of Mulch and Fertilization on the Quantity and Quality of Perennial Wall–Rocket (Diplotaxis tenuifolia)"

_plants, 2025, doi:10.3390/plants14101421_

Round 1
Reviewer 1 Report
Comments and Suggestions for Authors
General Comments
The manuscript provides valuable insights into the relationship between fertilization practices, effects of mulch, and the quantity and quality of perennial wall‒rocket. However, data interpretation, and contextualization require significant improvement. Below are specific concerns and recommendations.
Specific Comments
Abstract: Rewriting abstract based on results to be written using statistics
Introdaction: The introduction is well-structured and effectively covers all the topics addressed in the article.
AIM: To synthesize the aim of the study in a way that makes it more fluent and impactful.
Results
Line 112-116: In Figure 1, errors are found in the assignment of letters indicating statistical differences. It is not consistent that the WLDPE treatment, with higher values, is indicated with the letter "ab", while BLDPE, with lower values, is marked with the letter "a". This situation is not statistically plausible. It is assumed that there was an inversion: WLDPE should be indicated with the letter "a" and BLDPE with "ab". It is necessary to correct both the figure and the related comment, specifying that only the WLDPE treatment is statistically superior to the non-mulched control, while BLDPE does not show a significant difference.
Line 131: Check the percentage of difference because I can't find it correct
Line 155-167: The comment related to Figure 2 describes a yield trend among treatments that is not supported by the statistical results, as no significant differences were observed. Therefore, it is recommended to remove both the figure and the corresponding comment, as they may be misleading. It is preferable to refer exclusively to the data reported in Table 1, where the relevant numerical and statistical information is already provided.
Line 175-182: It is recommended to remove either "dry matter" or "water content" from the table, as they are complementary values (sum to 100%) and therefore redundant. Additionally, the statistical analysis should be checked, because it is inconsistent that the NM treatment is labeled as ab for water content, but as b for dry matter. Since these values are inversely related, the statistical groupings should reflect the same pattern, albeit in reverse.
Line 191-195: Since the data on TPC (total polyphenol content) and tannins do not show statistically significant differences, it is recommended not to include these data, as they do not provide meaningful information for the comparison among treatments.
Line 196-207: In this case as well, it is recommended not to include these data, as no statistically significant differences were observed among treatments. Including them would not provide meaningful insight into the results.
Line 2016-226: Regarding Table 3, it is also recommended to report only the data on β-carotene, as it is the only parameter showing statistically significant differences. The other data, which do not differ significantly among treatments, can be omitted to make the presentation clearer and more effective.
Line 240-241: It is not accurate to state that the lowest assimilation rate was observed only in the BLDPE × ChF treatment, as the NM × NF combination also showed similarly low values. The statement should be revised to accurately reflect the data.
Line 249-254: The comment should be revised, as it is not accurate to state that the highest number of leaves was recorded in the WLDPE × ChF combination, since this value is not statistically different from other treatments such as WLDPE × OF, WLDPE × NF, BLDPE × OF, etc.
Similar inconsistencies are present in the comments on other parameters as well, where differences are reported without statistical support. All comments should be carefully reviewed to ensure they reflect only statistically significant results.
Figura5: There are issues with the assignment of statistical letters in the figure. A thorough check of the statistical analysis is needed, and the group letters for each treatment should be corrected accordingly. Only after this verification should the comment be rewritten, ensuring that it reflects only statistically significant differences.
Line 280-284: In this table as well, it is advisable to report only one between water content (WC) or dry matter (DM), as they are complementary values. Additionally, the comment should be revised, paying close attention to the interpretation of statistical group letters: if one value is labeled "a" and another "ab", these two are not statistically different. The comment should be rewritten accordingly to avoid statements that are not supported by the statistical analysis.
Line 285-293: As for TPC (total polyphenol content), it is recommended not to report the data, since no statistically significant differences were observed among treatments.
Regarding tannins, the data should be interpreted with care, considering the correct meaning of the statistical group letters: values labeled with overlapping letters (e.g., "a" and "ab") are not statistically different.
In Figure 6, although differences in mean values between treatments are visible, only some of these differences are statistically significant. It is important to highlight that treatments sharing letters (e.g., "a" and "ab") are not statistically different. Therefore, the graph must be interpreted carefully, avoiding claims of superiority for treatments that are not statistically distinct. Make the comment again.
Table6: The comment related to Table 6 should be revised, as not all differences among treatments are statistically significant. In particular, treatments that share the same letter or partially overlapping letters (e.g., a and ab) are not statistically different. The revised comment should therefore focus only on the statistically significant differences, avoiding interpretations that are not supported by the analysis.
Line 406-416: This section should be moved to the introduction, as it provides general background information and is not directly supported by the results of the experiment. Therefore, it is not appropriate to keep it in the data discussion.
Discussion: Rewrite discussions based on the results to be written using statistics
Conclusion: Rewrite the coclusions based on the results to be written using statistics
Author Response
Dear Reviewer,
Thank you very much for your valuable recommendations and comments. We have carefully considered all your comments and recommendations and we have made changes in the manuscript.
All modifications were highlighted in red color.
- Abstract: Rewriting abstract based on results to be written using statistics
Answer: Thank you for your insightful comment. In response, the Abstract section has been thoroughly revised to better reflect the experimental findings by incorporating relevant statistical data.
- Introduction: The introduction is well-structured and effectively covers all the topics addressed in the article.
Answer: Thank you for your positive feedback. We appreciate your acknowledgment that the introduction is well-structured and successfully frames the scope of the study. This encourages us to maintain clarity and coherence throughout the manuscript.
- AIM: To synthesize the aim of the study in a way that makes it more fluent and impactful.
Answer: Thank you for your insightful comment. In response, the paper aim was change in: “In this context, this study aimed to evaluate the individual and combined effects of white polyethylene film (WLDPE), black polyethylene film (BLDPE), and non-mulched (NM) practices, in conjunction with organic (OF), chemical (ChF), and unfertilized fertilization (NF) regimes, on the yield and quality of the Bologna cultivar of perennial wall-rocket under the climatic conditions of northeastern Romania. To identify the optimal combination of factors for perennial wall-rocket, in the study were measured parameters such as COâ‚‚ assimilation rate, the number of leaves, the leaf area index per nest, yield, dry matter and water content, as well as antioxidant activities (measured via DPPH and ABTS assays), total phenolic content (TPC), tannin, chlorophyll A and B, lycopene, and β-carotene levels.”
- Line 112-116: In Figure 1, errors are found in the assignment of letters indicating statistical differences. It is not consistent that the WLDPE treatment, with higher values, is indicated with the letter "ab", while BLDPE, with lower values, is marked with the letter "a". This situation is not statistically plausible. It is assumed that there was an inversion: WLDPE should be indicated with the letter "a" and BLDPE with "ab". It is necessary to correct both the figure and the related comment, specifying that only the WLDPE treatment is statistically superior to the non-mulched control, while BLDPE does not show a significant difference.
Answer: Thank you for pointing out the inconsistency in the assignment of letters indicating statistical differences in Figure 1. Upon review, we agree that there was an inversion in the labeling of the WLDPE and BLDPE treatments. As suggested, we have corrected the figure and the related text. The WLDPE treatment is now correctly marked with the letter “a,” while BLDPE is marked with “ab.” Additionally, the revised text now clearly specifies that only the WLDPE treatment is statistically superior to the non-mulched control, whereas BLDPE does not show a significant difference.
We appreciate your attention to this detail and have made the necessary adjustments accordingly.
- Line 131: Check the percentage of difference because I can't find it correct
Answer: Thank you for pointing out the inconsistency, we have made the necessary adjustments accordingly.
- Line 155-167: The comment related to Figure 2 describes a yield trend among treatments that is not supported by the statistical results, as no significant differences were observed. Therefore, it is recommended to remove both the figure and the corresponding comment, as they may be misleading. It is preferable to refer exclusively to the data reported in Table 1, where the relevant numerical and statistical information is already provided.
Answer: We thank you for the comment regarding Figure 2 and the associated interpretation. While we acknowledge that no significant differences were observed among treatments in the first three harvests, we have chosen to retain Figure 2 because it provides relevant insights into yield dynamics across the harvesting period. Notably, significant differences did emerge in the fourth harvest, where the BLDPE mulch resulted in significantly lower yields compared to WLDPE, and chemical fertilization produced lower yields than both organic and unfertilized treatments. Furthermore, the data clearly show that the highest yields were obtained during the first two harvests, regardless of treatment, with a significant decline observed in the third and fourth.
- Line 175-182: It is recommended to remove either "dry matter" or "water content" from the table, as they are complementary values (sum to 100%) and therefore redundant. Additionally, the statistical analysis should be checked, because it is inconsistent that the NM treatment is labeled as ab for water content, but as b for dry matter. Since these values are inversely related, the statistical groupings should reflect the same pattern, albeit in reverse.
Answer: We appreciate your valuable suggestion. Upon reviewing the table, we have removed water content from the table to avoid redundancy. Furthermore, the PCA and Pearson correlation analyses have been redone to ensure accuracy and consistency with the revised dataset.
- Line 191-195: Since the data on TPC (total polyphenol content) and tannins do not show statistically significant differences, it is recommended not to include these data, as they do not provide meaningful information for the comparison among treatments.
Answer: We appreciate your observation regarding the lack of statistically significant differences in TPC and tannin content among treatments. While we acknowledge that these data do not contribute to treatment differentiation from a statistical standpoint, we have chosen to retain them in the manuscript for completeness and transparency. We believe this information may still be of interest to readers by providing a more comprehensive overview of the phytochemical profile and supporting the interpretation of treatment effects in a broader context. However, we have revised the text to clearly state that no significant differences were observed for these parameters.
- Line 196-207: In this case as well, it is recommended not to include these data, as no statistically significant differences were observed among treatments. Including them would not provide meaningful insight into the results.
Answer: We appreciate your observation regarding the lack of statistically significant differences but we have chosen to retain them in the manuscript for completeness and transparency. We believe this information may still be of interest to readers by providing a more comprehensive overview of the phytochemical profile and supporting the interpretation of treatment effects in a broader context. However, we have revised the text to clearly state that no significant differences were observed for these parameters.
- Line 2016-226: Regarding Table 3, it is also recommended to report only the data on β-carotene, as it is the only parameter showing statistically significant differences. The other data, which do not differ significantly among treatments, can be omitted to make the presentation clearer and more effective.
Answer: We thank you for the suggestion regarding Table 3. While it is true that some individual parameters did not show significant differences when analyzed independently, we have opted to retain all data in the table because the interaction between mulching and fertilization factors induced statistically significant differences. Therefore, we believe that including the full set of results provides a more accurate and comprehensive representation of the treatment effects. To enhance clarity, we have revised the corresponding text.
- Line 240-241: It is not accurate to state that the lowest assimilation rate was observed only in the BLDPE × ChF treatment, as the NM × NF combination also showed similarly low values. The statement should be revised to accurately reflect the data.
Answer: Thank you for the observation. We have revised the statement accordingly to accurately reflect the data.
- Line 249-254: The comment should be revised, as it is not accurate to state that the highest number of leaves was recorded in the WLDPE × ChF combination, since this value is not statistically different from other treatments such as WLDPE × OF, WLDPE × NF, BLDPE × OF, etc. Similar inconsistencies are present in the comments on other parameters as well, where differences are reported without statistical support. All comments should be carefully reviewed to ensure they reflect only statistically significant results.
Answer: Thank you for the observation. We have revised the comment and other related sections to reflect only statistically significant differences.
- Figure 5: There are issues with the assignment of statistical letters in the figure. A thorough check of the statistical analysis is needed, and the group letters for each treatment should be corrected accordingly. Only after this verification should the comment be rewritten, ensuring that it reflects only statistically significant differences.
Answer: Thank you for pointing this out. We have rechecked the statistical analysis for Figure 5 and corrected the group letters accordingly. The related comment has also been revised to accurately reflect only the statistically significant differences.
- Line 280-284: In this table as well, it is advisable to report only one between water content (WC) or dry matter (DM), as they are complementary values. Additionally, the comment should be revised, paying close attention to the interpretation of statistical group letters: if one value is labeled "a" and another "ab", these two are not statistically different. The comment should be rewritten accordingly to avoid statements that are not supported by the statistical analysis.
Answer: Thank you for the helpful observation. We have revised the table to report only one parameter (water content) to avoid redundancy. Additionally, the accompanying comment has been corrected to accurately reflect the statistical groupings and avoid any interpretation not supported by the analysis.
- Line 285-293: As for TPC (total polyphenol content), it is recommended not to report the data, since no statistically significant differences were observed among treatments.
Answer: Thank you for the suggestion. Although no statistically significant differences were observed for TPC, we have decided to retain the data in the manuscript for completeness and to support the interpretation of the overall phytochemical profile. However, we have clarified in the text that no significant differences were found.
- Regarding tannins, the data should be interpreted with care, considering the correct meaning of the statistical group letters: values labeled with overlapping letters (e.g., "a" and "ab") are not statistically different.
Answer: Thank you for the observation. We have revised the comment and other related sections to reflect only statistically significant differences.
- In Figure 6, although differences in mean values between treatments are visible, only some of these differences are statistically significant. It is important to highlight that treatments sharing letters (e.g., "a" and "ab") are not statistically different. Therefore, the graph must be interpreted carefully, avoiding claims of superiority for treatments that are not statistically distinct. Make the comment again.
Answer: Thank you for the observation. We have revised the comment and other related sections to reflect only statistically significant differences.
- Table 6: The comment related to Table 6 should be revised, as not all differences among treatments are statistically significant. In particular, treatments that share the same letter or partially overlapping letters (e.g., a and ab) are not statistically different. The revised comment should therefore focus only on the statistically significant differences, avoiding interpretations that are not supported by the analysis.
Answer: Thank you for the observation. We have revised the comment related to Table 6 to ensure that it reflects only statistically significant differences and avoids unsupported interpretations based on overlapping statistical groupings.
- Line 406-416: This section should be moved to the introduction, as it provides general background information and is not directly supported by the results of the experiment. Therefore, it is not appropriate to keep it in the data discussion.
Answer: Thank you for your helpful comment. We agree with your suggestion that this section would be more appropriate in the Introduction, as it provides general background information that supports the broader context of the study but is not directly linked to the results presented. In response, we have moved the relevant content in the first part of the Introduction section.
- Discussion: Rewrite discussions based on the results to be written using statistics
Answer: Thank you for your valuable suggestion. In response, we have rewritten the Discussion section to incorporate relevant statistical data that reflects the results more precisely.
- Conclusion: Rewrite the conclusions based on the results to be written using statistics
Answer: Thank you for your insightful comment. In response, the Conclusion section has been thoroughly revised to better reflect the experimental findings by incorporating relevant statistical data.

Reviewer 2 Report
Comments and Suggestions for Authors
- The specific control measures of temperature and humidity in the plastic greenhouse (such as whether ventilation or sunshade is used) are not clearly stated. Figure 11 only shows the trend of temperature and humidity, but does not analyze its potential impact on the results. It is necessary to supplement the correlation analysis between environmental factors and experimental treatment.
- "three independent replications" is mentioned in the text, but the specific sample size of each treatment (such as the number of plants per replicate) is not stated, which needs to be supplemented.
- The application amount of organic fertilizer and chemical fertilizer (e.g. 28 kg NPK vs. 200 kg organic fertilizer /1000 m ²) does not cite literature to support its rationality, which needs to be supplemented with calculation basis or compared with previous studies.
- The summary can supplement the guiding significance of the results for sustainable agriculture or regional crop management.
- The preface does not clearly point out that there are research gaps (such as the mechanism differences of photosynthetic pigment accumulation by different covering colors), so it is necessary to strengthen the research hypothesis at the end of the introduction.
- Some treatment combinations in Figure 4-6 do not show significant differences, and the possible reasons (such as insufficient sample size or environmental interference) need to be discussed.
- The description of PCA results in Figure 7-9 is more general, and it is suggested to explain the biological significance of principal components more specifically in combination with the load matrix (such as table 7).
8.The year of some references is inconsistent with the text citation
- In the discussion part, it is necessary to explain the physiological mechanism of white mulching film + organic fertilizer to improve yield (such as root temperature regulation and microbial activity enhancement).
- Language needs further polishing
Author Response
Dear Reviewer,
Thank you very much for your valuable recommendations and comments. We have carefully considered all your comments and recommendations and we have made changes in the manuscript.
All modifications were highlighted in red color.
- The specific control measures of temperature and humidity in the plastic greenhouse (such as whether ventilation or sunshade is used) are not clearly stated. Figure 11 only shows the trend of temperature and humidity, but does not analyze its potential impact on the results. It is necessary to supplement the correlation analysis between environmental factors and experimental treatment.
Answer: We appreciate your valuable observation and suggestion. At the time the experiments were conducted, it was not possible to record data on soil temperature and humidity at root level due to limitations in available equipment and monitoring infrastructure. As a result, only air temperature and relative humidity inside the greenhouse were tracked, as shown in Figure 11. These environmental parameters were monitored to ensure general uniformity across treatments, but a detailed analysis of their influence on the experimental outcomes could not be performed. We fully agree that soil microclimate plays a crucial role in plant physiological responses, and we recognize the value of such correlations. Therefore, we plan to incorporate this aspect into future research by implementing root-zone environmental monitoring and conducting statistical analyses to explore the relationship between environmental factors and plant performance.
- "three independent replications" is mentioned in the text, but the specific sample size of each treatment (such as the number of plants per replicate) is not stated, which needs to be supplemented.
Answer: In the Materials and Methods section, the following information was added: “Experiments were conducted in triplicate, with each replicate consisting of 12 nests. For each nest in an alveolar tray was sown 20 seeds and the seedlings were grown in greenhouse at 18–20°C day/16–18°C night and 70–75% relative humidity (RH).”
- The application amount of organic fertilizer and chemical fertilizer (e.g. 28 kg NPK vs. 200 kg organic fertilizer /1000 m ²) does not cite literature to support its rationality, which needs to be supplemented with calculation basis or compared with previous studies.
Answer: In the Materials and Methods section, the following information was added: “The fertilizer doses were determined in accordance with the provisions of EU Regulation 848/2018 on organic production, which permits the application of up to a maximum of 170 kg N/ha/year [72]. Furthermore, considering that only 60% of the nitrogen, phosphorus and potassium from organic fertilizer is available for plant uptake in the year of application [73] and that two rocket crops can be cultivated on the same plot within a single year, the fertilizer application rates were calculated accordingly.”
- The summary can supplement the guiding significance of the results for sustainable agriculture or regional crop management.
Answer: Thank you for the insightful suggestion. We have revised the Abstract and Conclusion sections to better highlight the practical implications of our findings for sustainable agriculture and regional crop management.
- The preface does not clearly point out that there are research gaps (such as the mechanism differences of photosynthetic pigment accumulation by different covering colors), so it is necessary to strengthen the research hypothesis at the end of the introduction.
Answer: Thank you for this valuable observation. To address your comment, the following paragraph was added: “The type of mulching and fertilization regime significantly can impact and the synthesis of both assimilatory pigments, such as chlorophyll a and b, lycopene, and carotenoids, as well as secondary metabolites like phenolic compounds and flavonoids [38–40], compounds that offer significant benefits both to plant health and in human nutrition [41,42]. According to the results of Caruso et al. [38] study, the variation in mulching type led to significant differences in the content of polyphenols, ascorbic acid, lipophilic antioxidant activity, hydrophilic antioxidant activity, or color components of perennial wall rocket. Similarly, Franquera [40] found that the levels of chlorophyll a and b in lettuce were significantly higher under yellow and red plastic mulch compared to silver, orange, and green mulch. In contrast, the red plastic mulch induced the highest content of total soluble solids, while the yellow plastic mulch led to the highest sugar content in lettuce. Regarding the fertilization regime, Keçe et al. [43] reported that the chlorophyll contents in lettuce leaves were substantially higher under the influence of organic and inorganic fertilization, compared to those detected in the non-fertilized variant. Instead, the study by da Cruz Bento et al. [44] showed that lettuce fertilized with conventional mineral fertilizer, tanned cattle manure, or Geofert organic-mineral fertilizer had lower chlorophyll a, but higher chlorophyll b content than non-fertilized plants.”
- Some treatment combinations in Figure 4-6 do not show significant differences, and the possible reasons (such as insufficient sample size or environmental interference) need to be discussed.
Answer: Thank you for this valuable observation. We acknowledge that some treatment combinations presented in Figures 4–6 did not show statistically significant differences. This may be attributed to factors such as limited sample size, natural biological variability, or environmental interference within the greenhouse environment. Although care was taken to maintain uniform conditions, subtle variations in microclimate or soil properties may have influenced plant responses. We have added a brief discussion on these aspects in the Discussion section to provide context for the observed variability.
- The description of PCA results in Figure 7-9 is more general, and it is suggested to explain the biological significance of principal components more specifically in combination with the load matrix (such as table 7).
Answer: Thank you for this valuable observation. To address your comment, the following changes were made: Biologically, the PCA demonstrates that mulching and fertilization regimes distinctly influence both agronomic performance and biochemical composition. Variants such as WLDPE × ChF clustered near productivity traits, while WLDPE × OF was associated with enhanced photosynthetic and antioxidant parameters. Conversely, BLDPE × NF was linked to increased pigment levels (chlorophyll B and β-carotene), and NM × NF, WLDPE × NF, and NM × ChF aligned with high levels of phenolic compounds and antioxidant capacity. These findings underscore how specific agronomic practices can modulate growth, pigment synthesis, and bioactive compound accumulation in perennial wall-rocket.
- The year of some references is inconsistent with the text citation
Answer: Thank you for pointing out the inconsistency regarding the publication years of some references. We have carefully reviewed the in-text citations and corresponding references, and we have corrected all discrepancies to ensure consistency between the citation years in the manuscript and those listed in the reference section.
- In the discussion part, it is necessary to explain the physiological mechanism of white mulching film + organic fertilizer to improve yield (such as root temperature regulation and microbial activity enhancement).
Answer: Thank you for this insightful suggestion. To address your comment, we have expanded the Discussion section to include a physiological explanation of how the combination of white mulching film and organic fertilization may contribute to improved yield. Specifically, we discussed how white mulch reflects sunlight, helping to regulate root-zone temperature and reduce heat stress, which can support optimal root development and nutrient uptake. Additionally, the application of organic fertilizer enhances soil microbial activity and improves soil structure, promoting better nutrient availability and root-soil interactions. These combined effects likely contributed to increased photosynthetic efficiency and biomass accumulation observed in the WLDPE × OF treatment.
- Language needs further polishing
Answer: We appreciate the reviewer’s comment regarding the language quality. The manuscript has been thoroughly revised to improve clarity, grammar, and overall readability. Particular attention was given to sentence structure, academic tone, and terminology consistency. We trust that the revised version reflects a higher standard of language and is now suitable for publication.

Round 2
Reviewer 1 Report
Comments and Suggestions for Authors
Great work. Now the manuscript is much clearer
Reviewer 2 Report
Comments and Suggestions for Authors
Accept in present form